# Performance Analysis of an IRS-Assisted SWIPT System with Phase Error and Interference

**DOI:** 10.3390/s25123756

**Published:** 2025-06-16

**Authors:** Xuhua Tian, Jing Guo, Zhili Ren

**Affiliations:** School of Electronic Information and Artificial Intelligence, Shaanxi University of Science and Technology, Xi’an 710021, China; x.h.tian@sust.edu.cn (X.T.); renzhili1999@163.com (Z.R.)

**Keywords:** IRS, SWIPT, interference, phase error, outage probability, ergodic capacity, energy efficiency

## Abstract

In this paper, we investigate a simultaneous wireless information and power transfer (SWIPT) communication system enhanced by an intelligent reflecting surface (IRS). Our study takes into account the imperfections in the phase shift of the IRS and the presence of interfering signals reflected by the IRS at the destination terminal. Additionally, our analysis incorporates both the presence of a line-of-sight path between the source and destination and a non-linear energy-harvesting model. In order to assess the influence of phase error and interference on the considered system, closed-form and asymptotic expression for the system’s outage probability, ergodic capacity, and energy efficiency (EE) are derived. Simulation results are presented to corroborate our analysis and illustrate the impact of phase error, interference, the number of reflecting elements, and various system parameters on the system performance.

## 1. Introduction

As a promising candidate for next-generation wireless systems, intelligent reflecting surfaces (IRSs) have garnered significant research interest owing to their low energy cost, low latency, and deployment feasibility. Their structure consists of many electronically controllable reflecting elements (RE) which can independently change the phase and/or amplitude of the incident signal [1]. With the aid of RE, IRSs can intelligently modify the propagation environment to improve the performance of the wireless system [2]. Due to their key benefits, IRSs have been widely exploited in various wireless communication scenarios, e.g., simultaneous wireless information and power transfer (SWIPT), which enables a time-switching (TS) or power-splitting (PS) receiver to harvest energy from the received signal so as to prolong the life of wireless networks [3]. The integration of IRSs into SWIPT systems offers several significant advantages, improving both the efficiency and performance of wireless communication and energy harvesting. An IRS-assisted SWIPT system was first investigated in [4], which established the fundamental framework for this emerging research area. By jointly optimizing the transmit precoders at the access point and the IRS phase shift matrix, the weighted sum power received by the energy-harvesting receivers is maximized. The proposed design demonstrated that the IRS-assisted SWIPT system can achieve an optimal rate-energy trade-off. Recent research on IRS-SWIPT systems has focused on addressing critical challenges such as optimal resource allocation, beamforming design, and energy efficiency maximization [5,6,7]. Studies have demonstrated that IRSs can significantly enhance the performance of SWIPT by creating favorable propagation environments through intelligent signal reflection and phase manipulation, thus improving both the energy-harvesting capability and the quality of information transmission [8,9]. Ref. [10] considered the resource allocation problem of minimizing energy consumption in an IRS-aided SWIPT mobile edge computing system, addressing the crucial aspect of energy efficiency in practical implementations.

Despite these advancements, significant challenges remain unresolved. The optimal IRS deployment in SWIPT systems, the fundamental trade-off between energy-harvesting and information-decoding performance, and the influence of practical constraints (e.g., hardware non-idealities and channel estimation inaccuracies) continue to be actively investigated [11]. Most works assume that the propagation channel through the IRS can be estimated perfectly and that the reflector phases induced by the IRS can then be set precisely. However, in practice, due to imperfect channel state information or low-precision phase shifts induced by the IRS, phase errors always exist. Several studies have analyzed the performance of IRS-SWIPT systems under phase errors. In [12], the authors analyzed the comprehensive performance of an SWIPT system with an IRS by adopting a hybrid TS-PS protocol. The impact of phase shift quantization and power allocation factors on the achievable rate and the energy harvested from the destination was investigated. The authors in [13] demonstrated that phase errors could lead to a significant drop in the signal-to-noise ratio (SNR) at the receiver, thereby reducing the efficiency of energy harvesting and information transmission. In [14], the authors derived closed-form expressions for both the achievable rate and harvested energy in the presence of phase errors, offering fundamental insights into the trade-offs between system performance and phase error tolerance.

Moreover, practical scenarios often involve interference, which can severely degrade system performance. However, the impact of co-channel interference (CCI) on IRS-assisted systems has not yet been widely studied. In scenarios where an excessive number of IRSs are deployed, particularly when the precise channel state information (CSI) of the interference is unavailable, the system may fail to suppress the undesired signals fully. Consequently, the residual interference caused by reflections cannot be ignored. In [15], the authors investigated an IRS-assisted interference-limited communication network in which only the destination was affected by CCI. The impact of interference on the system performance was analyzed in terms of outage probability, average bit error rate, and channel capacity. In [16], the authors focused on modeling the interference propagation via IRSs. The expression of a stochastic model for the interference level in RIS powered systems was derived. A downlink IRS-aided directional transmission network was considered in [17], where receivers experienced both the desired signal and IRS-reflected interference. Using stochastic geometry, closed-form expressions were derived for three key performance metrics: network coverage probability, spectral efficiency (SE), and energy efficiency (EE). In SWIPT-IRS systems, CCI can disrupt the precise beamforming and phase-shifting capabilities of the IRS, reducing the efficiency of both energy transfer and information transmission. The authors in [18] demonstrated that CCI can cause a substantial reduction in the harvested energy and achievable data rates, particularly in multiuser scenarios. In [19], the authors derived analytical expressions for the achievable rate and harvested energy in the presence of CCI, providing insights into the trade-offs between interference mitigation and system performance. Similarly, Ref. [20] investigated the impact of CCI on the outage probability and energy efficiency of SWIPT-IRS systems, highlighting the importance of robust interference management strategies.

### 1.1. Motivation and Contribution

Despite significant progress, the practical deployment of SWIPT-IRS systems remains challenging due to the dynamic nature of wireless environments and hardware limitations. Investigating the combined impact of CCI and phase errors on SWIPT IRS systems is essential for developing robust algorithms and strategies to mitigate these impairments. Ref. [21] investigated the joint impact of CCI and quantized phase shifts on an IRS-aided communication system by evaluating its coverage probability and ergodic capacity. The results showed that, compared to the ideal scenario, the presence of CCI and phase errors led to a severe deterioration in system performance. However, to the best of the authors’ knowledge, no existing works have investigated the combined impact of CCI and phase errors on an SWIPT IRS wireless system. Moreover, compared with direct link interference [22], the interference reflected by IRSs has not been studied in depth. Inspired by these considerations, in this paper, we investigate the performance of an IRS-assisted SWIPT system with phase errors and interference. The main contributions of our work are as follows:We present a model to characterize the impact of phase errors and interference on an IRS-aided SWIPT system. A source communicates with a destination through a direct link and an IRS, where the IRS reflects to the destination not only the desired signal but also the interference. The direct link from the interference to the destination is blocked due to obstacles. In particular, it is assumed that the phase errors for each reflecting element follow a von Mises distribution, which is also known as a circular normal distribution. Moreover, the TS-SWIPT scheme and non-linear energy harvesting (EH) model are applied.To investigate the impact of phase errors and interference, we present a comprehensive performance analysis by deriving closed-form and asymptotic expressions for the system outage probability, ergodic capacity, and EE, utilizing the obtained novel expression for the probability density function (PDF) and cumulative function (CDF) of the system end-to-end signal-to-interference ratio (SIR). By using the asymptotic outage performance in the high SNR regime, the system diversity order and coding gain are quantified. Moreover, the upper bounds of the system ergodic capacity are obtained as the number of reflecting elements N→∞.Finally, our analyzed expressions are verified by Monte Carlo simulations. Based on the theoretical analysis and simulation results, the effect of phase errors, interference, the number of reflecting elements of the IRS, and various system parameters on the IRS SWIPT system performance is discussed.

### 1.2. Notations and Organizations

A list of commonly used symbols in this paper is given in Table 1. The remainder of this paper is organized as follows:. Section 2 presents the system model for an IRS-assisted SWIPT system with phase errors and interference. The end-to-end SIR and energy-harvesting scheme is introduced. Section 3 derives analytical expressions for the outage probability, ergodic capacity, and energy efficiency. Section 4 validates the theoretical framework through numerical simulations. Finally, Section 5 concludes the paper with key findings and future research directions.

## 2. System Model

Consider a downlink communication network where a source terminal S and a destination terminal D communicate with each other via the direct link and an IRS with *N* reflecting elements (REs) $R_{n}$, n=1,2,…,N. It is assumed that the IRS is affected by *L* independent interfering signals $I_{l}$, l=1,2,…,L, as shown in Figure 1. This scenario happens when the IRS is located at the edge of a cellular network. Although the direct link between the interfering signal and the destination is blocked, the interfering signal can be easily reflected by the IRS. $h_{\mathrm{SR}}=\left[h_{{\mathrm{SR}}_{1}},\ldots ,h_{{\mathrm{SR}}_{N}}\right]\in C^{N\times 1}$, $h_{\mathrm{RD}}=\left[h_{R_{1}D},\ldots ,h_{R_{N}D}\right]\in C^{1\times N}$, $h_{I_{l}R}=\left[h_{I_{l}R_{1}},\ldots ,h_{I_{l}R_{N}}\right]\in C^{N\times 1}$, and $h_{\mathrm{SD}}\in C^{1\times 1}$ are the channel gains for the S-to-IRS, IRS-to-D, $I_{l}$-to-IRS, and S-to-D links, respectively, where $C^{x\times y}$ represents the space of x×y complex-valued matrices. We have $h_{{\mathrm{SR}}_{n}}=d_{{\mathrm{SR}}_{n}}^{-\frac{v}{2}}\alpha_{{\mathrm{SR}}_{n}}e^{-j\theta_{{\mathrm{SR}}_{n}}}$, $h_{R_{n}D}=d_{R_{n}D}^{-\frac{v}{2}}\alpha_{R_{n}D}e^{-j\theta_{R_{n}D}}$, $h_{I_{l}R_{n}}=d_{I_{l}R_{n}}^{-\frac{v}{2}}\alpha_{I_{l}R_{n}}e^{-j\theta_{I_{l}R_{n}}}$, and $h_{\mathrm{SD}}=d_{\mathrm{SD}}^{-\frac{v}{2}}\alpha_{\mathrm{SD}}e^{-j\theta_{\mathrm{SD}}}$, where $\alpha_{{\mathrm{SR}}_{n}}$, $\alpha_{R_{n}D}$, $\alpha_{I_{l}R_{n}}$, and $\alpha_{\mathrm{SD}}$ are the channel’s amplitudes, $\theta_{\mathrm{AB}}$ is the phase shift, $d_{\mathrm{AB}}$ denotes the lengths of the A-B link, and *v* is the path loss exponent. We assume that the distance of the IRS-D link is much longer than the height of the IRS above the ground. Hence, the height of the IRS is ignored. Hence, $d_{{\mathrm{SR}}_{n}}=d_{\mathrm{SR}}$, $d_{R_{n}D}=d_{\mathrm{RD}}$, and $d_{I_{l}R_{n}}=d_{I_{l}R}$ for n=1,2,…N. All fading channels $h_{{\mathrm{SR}}_{n}}∼\mathrm{CN}\left(0,d_{\mathrm{SR}}^{-v}\right)$, $h_{R_{n}D}∼\mathrm{CN}\left(0,d_{\mathrm{RD}}^{-v}\right)$, $h_{I_{l}R_{n}}∼\mathrm{CN}\left(0,d_{I_{l}R}^{-v}\right)$, and $h_{\mathrm{SD}}∼\mathrm{CN}\left(0,d_{\mathrm{SD}}^{-v}\right)$ are assumed to be independent and identically distributed (i.i.d.) and follow the Rayleigh distribution due to the isotropic scattering assumption (e.g., urban or indoor environments with dense obstructions [23,24]).

Let $x_{s}$ and $x_{l}$ be the unit norm symbols transmitted by S and $I_{l}$, respectively. The received signal at D can be written as(1)$$\begin{array}{c} y=\sqrt{P_{S}}\left[\sum_{n=1}^{N}h_{R_{n}D}e^{j\theta_{n}}h_{{\mathrm{SR}}_{n}}+h_{\mathrm{SD}}\right]x_{s}+\sum_{l=1}^{L}\left[\sum_{n=1}^{N}\sqrt{P_{I_{l}}}h_{R_{n}D}e^{j\theta_{n}}h_{I_{l}R_{n}}\right]x_{l}+n_{d}, \end{array}$$
where $P_{S}$ and $P_{I_{l}}$ are the average power transmitted by S and $I_{l}$. $\theta_{n}$, n=1,2,…,N represents the phase shift induced by the *n*th RE of the IRS. $n_{d}∼\mathrm{CN}\left(0,1\right)$ is the additive white Gaussian noise (AWGN).

Since the IRS can also reflect signals from co-channel interference, the system suffers from many interferers. Accordingly, we assume that the considered network operates in an interference-limited regime [25,26]. The received SIR can be written as(2)$$\gamma =\frac{P_{S}{\left|\sum_{n=1}^{N}d_{\mathrm{SR}}^{-\frac{v}{2}}d_{RD}^{-\frac{v}{2}}\left|\alpha_{{\mathrm{SR}}_{n}}\right|\left|\alpha_{R_{n}D}\right|e^{j\delta_{n}}+d_{\mathrm{SD}}^{-\frac{v}{2}}\alpha_{\mathrm{SD}}\right|}^{2}}{\sum_{l=1}^{L}\rho_{I_{l}}{\left|\sum_{n=1}^{N}\left|\alpha_{I_{l}R_{n}}\right|\left|\alpha_{R_{n}D}\right|e^{jv_{ln}}\right|}^{2}},$$
where $\rho_{I_{l}}=P_{I_{l}}/d_{I_{l}R_{n}}^{-v}d_{R_{n}D}^{-v}$, $\delta_{n}=\theta_{n}-\theta_{{\mathrm{SR}}_{n}}-\theta_{R_{n}D}$, and $v_{ln}=\theta_{n}-\theta_{I_{l}R_{n}}-\theta_{R_{n}D}$. Ideally, assuming perfect knowledge of all the channels’ phases, the *n*th RE can choose a phase shift to maximize the received SINR at D (i.e., $\theta_{n}=\theta_{{\mathrm{SR}}_{n}}+\theta_{R_{n}D}$). However, in practice, due to imperfect CSI and finite precision in phase adjustment, a random phase error still persists. Here, we assume the phase error $\delta_{n}$ is independent, and identical von Mises random variables (RVs) with a zero mean and concentration parameter κ [27,28] are the probability density function (PDF) given by(3)$$f_{\delta_{n}}\left(x\right)=\frac{e^{\kappa \cos(x)}}{2\pi I_{0}\left(\kappa \right)},\;-\pi <x<\pi ,$$
where $I_{0}\left(\cdot \right)$ is the modified Bessel function of the first kind of zero order. Moreover, because we do not have any channel knowledge of the interference, and the phase shifts $\theta_{n}$ are provided to compensate for the effect of $S\to {\mathrm{RE}}_{n}$ and ${\mathrm{RE}}_{n}\to D$ links, we assume that $v_{ln}$ is uniformly distributed within [−π,π).

For the TS-based SWIPT system under consideration, during each channel coherence interval $T_{c}$, the received signal is split into two portions. The $\varepsilon T_{c}$ portion is used for energy harvesting (EH), and the remaining $\left(1-\varepsilon \right)T_{c}$ portion is used for information decoding, where ε is the sharing factor. Then, by considering a non-linear EH model [29] based on real measurements, the harvested energy can be quantified as(4)$$\mathrm{EH}=\frac{\varphi }{W\left(1+\exp\left(-a\left(P_{r}-b\right)\right)\right)}-Z,$$
where *a* and *b* are constants related to the circuit specifications, φ is the maximum output DC power, $W=\exp\left(ab\right)/\left(1+\exp(ab)\right)$, Z=φ/exp(ab), and the received energy $P_{r}$ is given by(5)$$\begin{array}{c} P_{r}=\varepsilon T_{c}\left(P_{S}{\left|\sum_{n=1}^{N}d_{\mathrm{SR}}^{-\frac{v}{2}}d_{RD}^{-\frac{v}{2}}\left|\alpha_{{\mathrm{SR}}_{n}}\right|\left|\alpha_{R_{n}D}\right|e^{j\delta_{n}}+d_{\mathrm{SD}}^{-\frac{v}{2}}\alpha_{\mathrm{SD}}\right|}^{2}+\sum_{l=1}^{L}\rho_{I_{l}}{\left|\sum_{n=1}^{N}\left|\alpha_{I_{l}R_{n}}\right|\left|\alpha_{R_{n}D}\right|e^{jv_{ln}}\right|}^{2}\right). \end{array}$$

## 3. Performance Analysis

In this section, we provide a theoretical analysis of the impact of phase errors and interference on the performance of the IRS-assisted SWIPT system. We evaluate the expression of the PDF and CDF for the end-to-end SIR, which is further utilized to derive the closed-form expressions of the outage probability and ergodic capacity. Specifically, the system diversity order and coding gain are formulated by using the asymptotic expressions of the outage probability, and the upper bounds of the system ergodic capacity are presented for N→∞. Finally, the energy efficiency based on the total power modeling is evaluated.

### 3.1. Outage Probability

Outage probability is defined as when the instantaneous receiver SIR falls below a given threshold $\gamma_{\mathrm{th}}$ and can be mathematically written as(6)$$P_{\mathrm{out}}=\mathrm{Pr}\left(\gamma <\gamma_{\mathrm{th}}\right),$$
where Pr(·) represents the probability.

In order to evaluate (6), we set $X=P_{S}{\left|\sum_{n=1}^{N}d_{\mathrm{SR}}^{-\frac{v}{2}}d_{RD}^{-\frac{v}{2}}\left|\alpha_{{\mathrm{SR}}_{n}}\right|\left|\alpha_{R_{n}D}\right|e^{j\delta_{n}}+d_{\mathrm{SD}}^{-\frac{v}{2}}\alpha_{\mathrm{SD}}\right|}^{2}$ and $Y=\sum_{l=1}^{L}\rho_{I_{l}}{\left|\sum_{n=1}^{N}\left|\alpha_{I_{l}R_{n}}\right|\left|\alpha_{R_{n}D}\right|e^{jv_{ln}}\right|}^{2}$. The PDFs $f_{X}\left(x\right)$ and $f_{Y}\left(y\right)$ for *X* and *Y* are required.

As shown in [30], the probability distributions of *X* and *Y* can be well approximated by Laguerre expansion, even for a small number of REs. This method is widely adopted in the literature (e.g., [31,32]) for analyzing composite fading channels. Different from [32], in our system model, the composite channel *X* contains the phase errors. The von Mises distribution for $\delta_{n}$ is decoupled from the Rayleigh fading assumption, as it models IRS phase quantization errors controlled by the IRS controller not the propagation environment. Therefore, using the Laguerre expansion to approximate *X* is also accurate. We verify the accuracy through Monte Carlo the simulation described in Section 4. Therefore, their PDFs can be expressed, respectively, as(7)$$f_{X}\left(x\right)=\frac{x^{m}}{n^{m+1}\Gamma \left(m+1\right)}\exp\left(-\frac{x}{n}\right),$$
and(8)$$f_{Y}\left(y\right)=\frac{y^{m_{I}}}{n_{I}^{m_{I}+1}\Gamma \left(m_{I}+1\right)}\exp\left(-\frac{y}{n_{I}}\right),$$
where Γ(·) is the gamma function [33] (8.310), $n=\sigma_{X}^{2}/\mu_{X}$, $m=\mu_{X}^{2}/\sigma_{X}^{2}-1$, $n_{I}=\sigma_{Y}^{2}/\mu_{Y}$, and $m_{I}=\mu_{Y}^{2}/\sigma_{Y}^{2}-1$. $\mu_{X}$, $\mu_{Y}$, $\sigma_{X}^{2}$, and $\sigma_{Y}^{2}$ are the means and variances of *X* and *Y*, respectively. The derivations are given in Appendix A.

Then, using [33] (3.326.2), the PDF of γ can be derived as(9)$$\begin{array}{cc} f_{\gamma }\left(z\right)= & \int_{0}^{\infty }yf_{X}\left(yz\right)f_{Y}\left(y\right)dy\\ = & \int_{0}^{\infty }\frac{z^{m}y^{m+m_{I}+1}\exp\left(-\left(\frac{z}{n}+\frac{1}{n_{I}}\right)y\right)}{n^{m+1}n_{I}^{m_{I}+1}\gamma \left(m+1\right)\gamma \left(m_{I}+1\right)}dy\\ = & \frac{{\left(\frac{n_{I}}{n}\right)}^{m+1}}{B\left(m+1,m_{I}+1\right)}\frac{z^{m}}{{\left(1+\frac{n_{I}}{n}z\right)}^{m+m_{I}+2}}, \end{array}$$
where B(·,·) is the beta function [33].

Furthermore, using [33] (3.194.1), the cumulative distribution function (CDF) of γ can be given by(10)Fγ(z)=zm+1Bm+1,mI+1(m+1)nInm+1F12m+mI+2,m+1;m+2;−nInz,
where Fqpa1,···,ai;b1,···,bj;z is the generalized hypergeometric function [33].

By inserting $z=\gamma_{\mathrm{th}}$ into (10), we can obtain the outage probability expression as(11)$$P_{\mathrm{out}}=F_{\gamma }\left(\gamma_{\mathrm{th}}\right).$$

To show the explicit impact of the system parameters on the SIR at D, we also provide an asymptotic outage analysis. In the high-power region, i.e., $\rho_{S}\to \infty$, limn→∞F12m+mI+2,m+1;m+2;−nInγth=1. Consequently, $P_{\mathrm{out}}$ can be asymptotically written as(12)$$\begin{array}{c} P_{\mathrm{out}}^{\infty }\approx \frac{1}{B\left(m+1,m_{I}+1\right)\left(m+1\right)}{\left(\frac{n_{I}\gamma_{\mathrm{th}}}{n}\right)}^{m+1}. \end{array}$$

From (12), it can be seen that the system diversity order is m+1, which is proportional to *N* and the phase errors. As expected, the system performance becomes better with the increase in $\rho_{S}$, but becomes worse with the increase in $\rho_{I}$ or the number of interferences *L*.

### 3.2. Ergodic Capacity

In this section, we present the closed-form and approximate expressions for the system ergodic capacity. By definition, first, we obtain the average ergodic capacity by using the Mellin transform [34,35] (2.31):(13)$$\begin{array}{cc} C= & \left(1-\varepsilon \right)T_{c}E\left[\log_{2}\left(1+\gamma \right)\right]\\ = & \left(1-\varepsilon \right)T_{c}\int_{0}^{\infty }\log_{2}\left(1+z\right)f_{\gamma }\left(z\right)dz\\ = & \frac{\left(1-\varepsilon \right)T_{c}}{2\pi j\ln2}\int_{\sigma -j\infty }^{\sigma +j\infty }f_{\gamma }^{*}\left(s\right)c^{*}\left(1-s\right)ds, \end{array}$$
where $c^{*}\left(s\right)=\frac{\Gamma (1+s)\Gamma (-s)}{-s}$ is the Mellin transform of ln(1+γ), $f_{\gamma }^{*}\left(s\right)$ is the Mellin transform of $f_{\gamma }\left(z\right)$, which can be expressed as(14)$$\begin{array}{cc} f_{\gamma }^{*}\left(s\right) & =\int_{0}^{\infty }f_{\gamma }\left(z\right)z^{s-1}dz\\ & ={\left(\frac{n_{I}}{n}\right)}^{1-s}\frac{\Gamma \left(m+s\right)\Gamma \left(2+m_{I}-s\right)}{\Gamma \left(m+1\right)\Gamma \left(m_{I}+1\right)}, \end{array}$$
and σ lies in the region of convergence of both $f_{\gamma }^{*}\left(s\right)$ and $c^{*}\left(1-s\right)$, $1<\sigma <\min\{2,m_{I}+2\}$. E[·] is the expectation operation.

Hence, using the definition of the Meijer G-function [35], the average ergodic capacity can be obtained as(15)$$C\;=\;\frac{\left(1\;-\;\varepsilon \right)T_{c}n}{\lambda \;\ln\;2n_{I}}G_{5,4}^{4,4}\;\left(\;\frac{n}{n_{I}}|\begin{array}{c} -m,-m_{I}-2,0,0,2\\ 0,0,1,1 \end{array}\;\right),$$
where $\lambda =\Gamma \left(m+1\right)\Gamma \left(m_{I}+1\right)$.

From this expression, it is difficult to obtain insightful observations. Applying Taylor’s expansion and the expectation operator, an approximation for *C* can be given by(16)$$C\approx \frac{\left(1\;-\;\varepsilon \right)T_{c}}{\ln2}\left[\ln\left(1+E\left[\gamma \right]\right)-\frac{E\left[\gamma^{2}\right]-E{\left[\gamma \right]}^{2}}{2{\left(1+E\left[\gamma \right]\right)}^{2}}\right].$$

According to [36], and applying the above derivation results, we have(17)$$\begin{array}{c} E\left[\gamma \right]=\frac{\mu_{X}}{\mu_{Y}}+\frac{\mu_{X}\sigma_{Y}^{2}}{\mu_{Y}^{3}}, \end{array}$$(18)$$\begin{array}{c} E\left[\gamma^{2}\right]=\frac{\sigma_{X}^{2}+\mu_{X}^{2}}{\sigma_{Y}^{2}+\mu_{Y}^{2}}+\frac{2\sigma_{Y}^{2}\left(\sigma_{X}^{2}+\mu_{X}^{2}\right)\left(\sigma_{Y}^{2}+2\mu_{Y}^{2}\right)}{{\left(\sigma_{Y}^{2}+\mu_{Y}^{2}\right)}^{3}}. \end{array}$$

Finally, by substituting (17) and (18) into (16), the closed form for the average capacity can be obtained. In particular, by setting N→∞, we can further bound the ergodic capacity as(19)$$\begin{array}{c} C∼\frac{\left(1\;-\;\varepsilon \right)T_{c}}{\ln2}\ln\left(1+\frac{\rho_{\;S}N\varphi^{2}}{\sum_{l=1}^{L}\rho_{I_{l}}}+\;\frac{\rho_{\;S}N\varphi^{2}\sum_{l=1}^{L}\rho_{I_{l}}^{2}}{{\left(\sum_{l=1}^{L}\rho_{I_{l}}\right)}^{3}}\right). \end{array}$$

### 3.3. Energy Efficiency

In this section, we present an analysis of the EE, which is an important metric for determining the sustainability of wireless networks. The EE is defined as the ratio of the system throughput to the total power consumption.

Firstly, the average harvested energy $P_{r}$ can be derived as(20)$$\begin{array}{cc} P_{r}= & \varepsilon T_{c}\eta E\left(X+Y\right)\\ = & \varepsilon T_{c}\eta \left(\mu_{x}+\mu_{Y}\right)\\ = & \varepsilon T_{c}\eta \left[\rho_{\;S}\left(1+N+N\left(N-1\right)\varphi^{2}\right)+\sum_{l=1}^{L}\rho_{I_{l}}N\right]. \end{array}$$

Following [29], the total power consumption is quantified as(21)$$P_{\mathrm{tot}}=w_{\max}P_{S}+P_{\mathrm{Tx}}+P_{\mathrm{Rx}}+NP_{n}-\mathrm{EH},$$
where $w_{\max}$ is the maximum efficiency of the transmit power $P_{S}$ amplifier, and $P_{\mathrm{Tx}}$ and $P_{\mathrm{Rx}}$ denote the total hardware dissipated static power at S and D, respectively. $P_{n}$ represents the power consumption of each RE.

Now, combining (21) together with (15) or (16), the EE can be obtained as(22)$$\eta_{EE}=\frac{C}{P_{\mathrm{tot}}}.$$

Note that system capacity no longer increases linearly with N. Instead, it is constrained by interference signals and phase error, resulting in a slower growth rate. Moreover, based on (22), it is clear that all the variables exhibit coupling characteristics. Simply increasing *N* or reducing interference signals is insufficient to improve the energy efficiency of the system. The detailed effects of these factors on the energy efficiency are provided in Section 4.

## 4. Simulation Results and Discussions

In this section, the simulation results are presented to verify our analysis and to evaluate the impact of phase error, interference, and various parameters on the system performance. During the simulations, we assume the number of interferences L=2 and $\rho_{I_{l}}=\rho_{I_{2}}=\rho_{I}$. The distances of the S-IRS, the IRS-D, and the S-D links are normalized to unity, i.e., $d_{\mathrm{SR}}=d_{\mathrm{RD}}=d_{\mathrm{SD}}=1$ [37]. The path loss exponent is v=2. Other default simulation parameters are as follows: $P_{\mathrm{Tx}}=P_{\mathrm{Rx}}=0.1$ W, $P_{n}=0.01$ W, $w_{\max}=1.2$, a=150, b=0.024, and φ=24 mW [38]. We further set the outage threshold $\gamma_{\mathrm{th}}$ as equal to 0 dB and the energy conversion efficiency as η=0.4.

In Figure 2, we compare the PDF of *X* obtained by the simulation and our approximation in (7). We set $P_{S}=0$ dBm. It shows that our approximation matches closely the PDF obtained by the simulation for different cases of *N* and κ, which implies that our PDF approximation is accurate. It can be seen that, for the same κ=2, the accuracy of the approximation is significantly improved by increasing *N* from 6 to 16. Similarly, for the same N=6, the accuracy of the approximation can be significantly improved by increasing κ from 2 to 8. It is observed that the accuracy of the approximation mainly depends on *N* and κ, i.e., having an N>6 and a κ>2. Moreover, it is worth mentioning that the PDF of *X* shifts to the right when *N* and κ are increased, which, in turn, enhances the gain of the S-to-D composite channel.

In Figure 3, the outage probability of the considered system is shown for the different values of κ, *N*, and $\rho_{I}$. From the figure, one can see that there is a large performance gap between the ideal case (no phase error and interference) and the practical case. As κ increases (for which the phase shift becomes more accurate), the outage performance can be greatly improved and the system diversity order is also increased. We can see that increasing $\rho_{I}$ leads to a severe deterioration of the system performance, but it cannot affect the system diversity order. Particularly, observing the two scenarios where N=8 and $\rho_{I}=0$ dB and N=16 and $\rho_{I}=10$ dB, in the low-power region, even as *N* increases, the $P_{\mathrm{out}}$ for $\rho_{I}=0$ dB surpasses that for $\rho_{I}=10$ dB. This is because the impact of reflected interference signals becomes more severe when *N* grows large. However, in the high-power region, the $P_{\mathrm{out}}$ for $\rho_{I}=10$ dB is lower. This is because the system’s diversity gain is drastically increased by increasing *N*.

In Figure 4, we present the impact of the number of REs *N*, κ, and $\rho_{I}$ on the ergodic capacity with increasing $P_{S}$, respectively. We can see that the ergodic capacity is increased by increasing the number of REs *N* or by increasing $\rho_{S}$. Moreover, it can be seen that, for N=16 and $\rho_{I}=0$ dB at $P_{S}=20$ dBm, the ergodic capacity reduces from 9 (bits/s/Hz) to 8 (bits/s/Hz) as κ decreases. However, for N=16 and κ=8 at $P_{S}=20$ dBm, the ergodic capacity drastically reduces from 9 to 5.8 when the interference increases from 0 dB to 10 dB. One can observe that reducing the interference can greatly improve system performance.

Figure 5 and Figure 6 show the curves of the harvested power under the non-linear EH model versus the number of REs *N* and the transmitted power $P_{S}$ with different κ and $\rho_{I}$ values at $P_{S}=10$ dBm and N=16, respectively. We can see that, with the increase in *N* or $P_{S}$, the harvested power grows non-linearly. From Figure 5, it can be seen that, when *N* is small, i.e., N<20, the harvested power is lower and increases relatively slowly. When 20<N<50, the harvested power grows rapidly as *N* increases, whereas, when *N* is larger, the harvested power eventually tends to saturate. It is obvious that, as the interference signal power increases, additional energy to the destination can be provided, but only a small amount of energy is harvested. On the other hand, as κ increases, more power is harvested. From Figure 6, it can be seen that, when $P_{S}$ is small, i.e., $P_{S}<10$ dBm, the harvested power is lower and increases relatively slowly. When 10 dBm $<P_{S}<20$ dBm, the harvested power grows rapidly as $P_{S}$ increases, whereas, when $P_{S}$ is larger, harvested power tends to saturate. Similarly, the harvested energy increases significantly with the increase in κ, while the impact of interference signals remains relatively small.

We illustrate the effect of phase error and interference on the EE performance versus the number of REs *N* and the transmitted power $P_{S}$ in Figure 7 and Figure 8, respectively. We consider two limitations of the energy-harvesting circuit: Firstly, the harvested energy tends to saturate as *N* and the received power grow larger. Secondly, when the received power falls below the EH circuit’s processing threshold, the harvested energy is zero. Therefore, we set the time allocation factor ε to ensure that ${\mathrm{EH}}_{\max}=24$ mW and ε=0 because the channel conditions are poor. In Figure 7, $P_{S}=20$ dBm. It can be seen that the EE first increases and then decreases as *N* increases. This is because, when *N* is relatively small, much performance gain is achieved. However, as *N* grows larger, although SE is improved, the power consumed also rises accordingly, leading to a reduction in EE. Furthermore, as $\rho_{I}$ increases, even though the interfering signals can provide additional harvested energy, their impact on SE is more pronounced. Consequently, the system’s energy efficiency with $\rho_{I}=0$ dB surpasses that with $\rho_{I}=10$ dB.

In Figure 8, N=16. It shows that, with an increase in $P_{S}$, EE first increases and then decreases, which means that merely increasing the transmission power does not effectively enhance the system’s energy efficiency. Similarly, we can observe that, compared to the phase errors, the impact of interfering signals on EE is more severe. Although the EE at $\rho_{I}=0$ dB is superior to that at $\rho_{I}=10$ dB, it declines at a faster rate. This is because, when $\rho_{I}=10$ dB, interfering signals affect the system’s transmission rate, and they also provide additional energy for harvesting, which can reduce the time factor ε required for energy collection and better serve the information transmission process.

Figure 9 plots the EE versus the number of IRS reflected elements *N* and the transmit power $P_{S}$. For clarity in trend observation, we set κ=8 and $\rho_{I}=0$ dB. Within the reasonable range of the parameters, we can see that there exists an optimal combination of *N* and $P_{S}$ to maximize EE. As can be observed in Figure 6 and Figure 7, EE does not continuously increase linearly with *N* and $P_{S}$. The optimal EE occurs at a moderate *N* (40–60) and moderate-to-high $P_{S}$ (15–25 dBm), where IRS-aided beamforming and power gains balance power consumption. This highlights the importance of joint optimization of *N* and transmit power in practical IRS deployments.

## 5. Conclusions

In this paper, we conducted a performance analysis of an RIS-based SWIPT system, taking into account phase errors and interference, through both theoretical derivation and simulation. Expressions for outage probability, ergodic capacity, and EE were derived. Our findings indicate that enhancing the precision of phase shifts and mitigating interference can significantly improve the system performance. The diversity order of the system in the presence of phase errors is reduced to m+1, whereas it remains at *N* in the ideal scenario. The reflection interference diminishes the coding gain of the system and exerts a considerable influence on both the ergodic capacity and the energy efficiency of the system. In addition, since the harvested energy attains a saturation point with the increase in the number of reflecting elements *N* and $\rho_{S}$, it consequently leads to a degradation in energy efficiency. Our analysis focused on static scenarios with fixed parameters. In future work, the implementation of the system will be considered to investigate the impact of mobility-induced Doppler shifts, carrier frequency selection, and bandwidth scalability on the performance of IRS-aided SWIPT systems. These extensions will enhance the robustness and scalability of IRS-SWIPT systems in real-world dynamic environments, paving the way for standardized deployments in 6G networks.

## Figures and Tables

**Figure 1 sensors-25-03756-f001:**
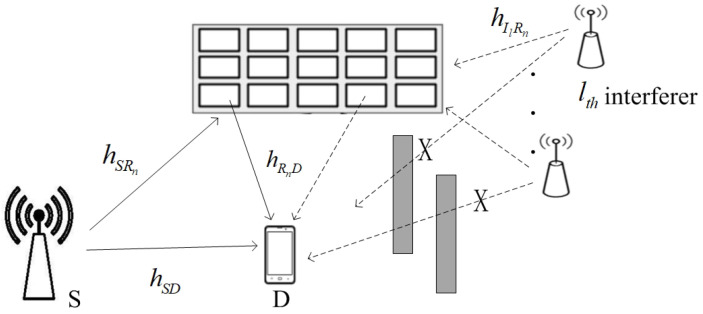
System model of an IRS-aided SWIPT wireless communication network in the presence of interferers. The symbol *X* indicates the absence of a direct link between the interference and D.

**Figure 2 sensors-25-03756-f002:**
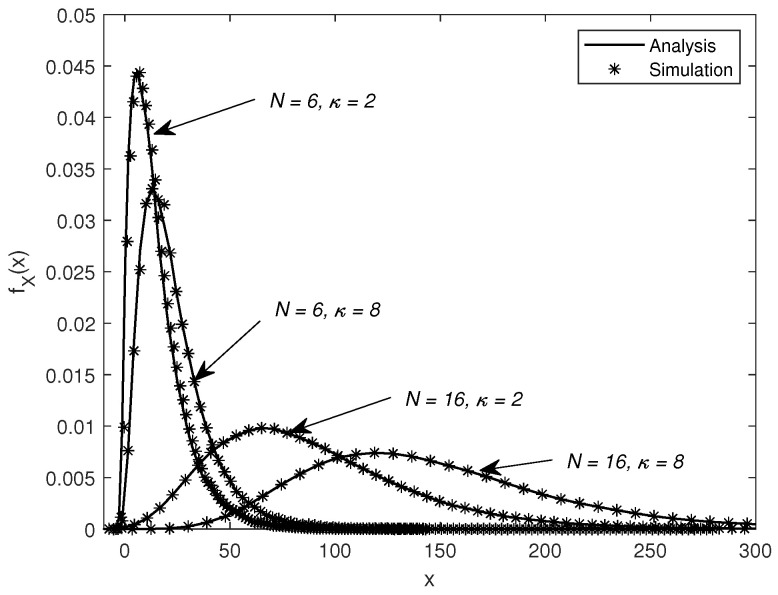
The PDF of X for different *N* and κ values.

**Figure 3 sensors-25-03756-f003:**
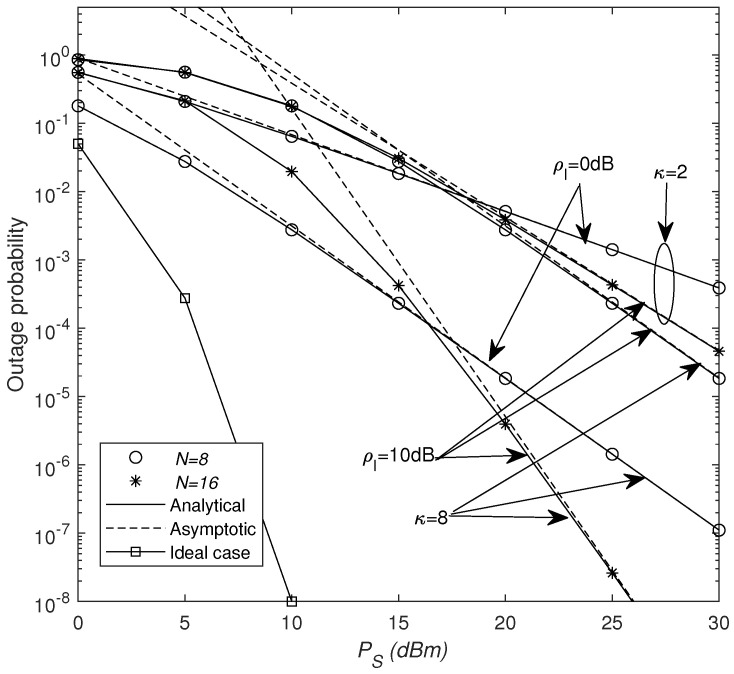
Outage probability versus $P_{S}$ for different κ and $\rho_{I}$ values.

**Figure 4 sensors-25-03756-f004:**
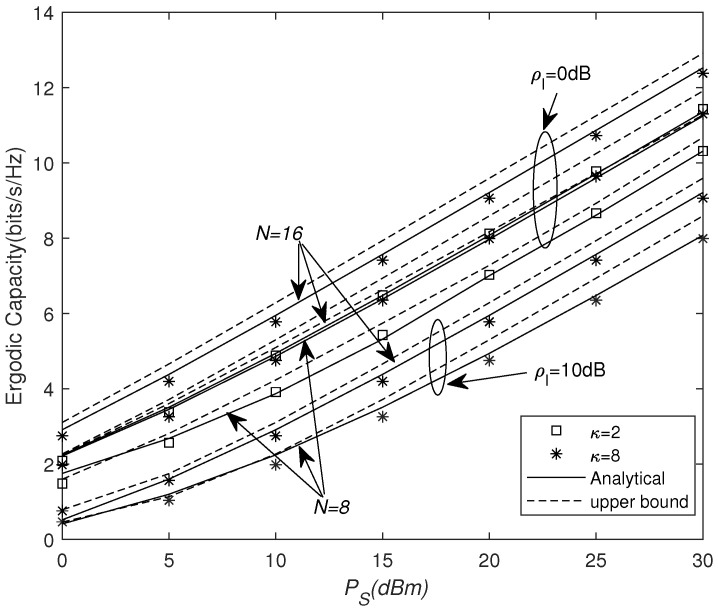
Spectral efficiency versus $P_{S}$ for different *N* and $\rho_{I}$ values.

**Figure 5 sensors-25-03756-f005:**
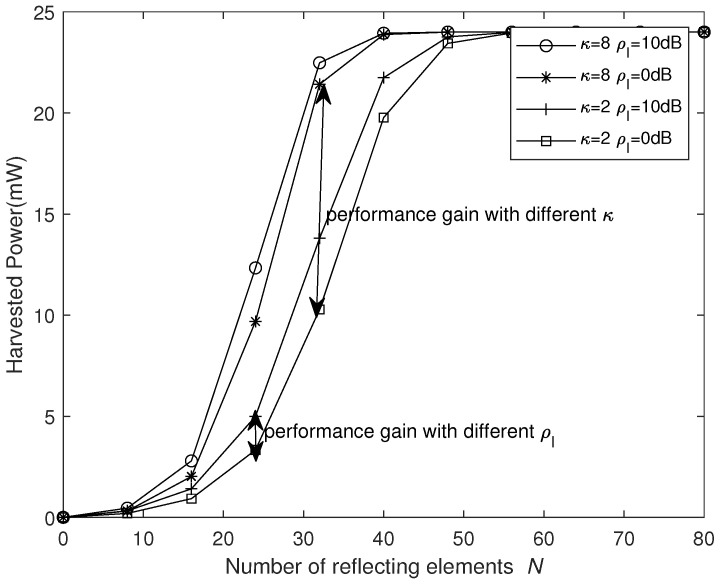
Harvested power versus *N* for different κ and $\rho_{I}$ values.

**Figure 6 sensors-25-03756-f006:**
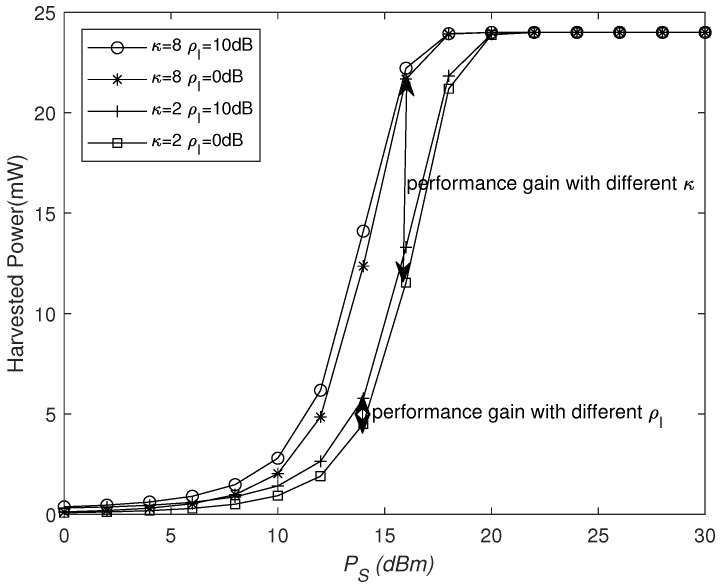
Harvested power versus $P_{S}$ for different κ and $\rho_{I}$ values.

**Figure 7 sensors-25-03756-f007:**
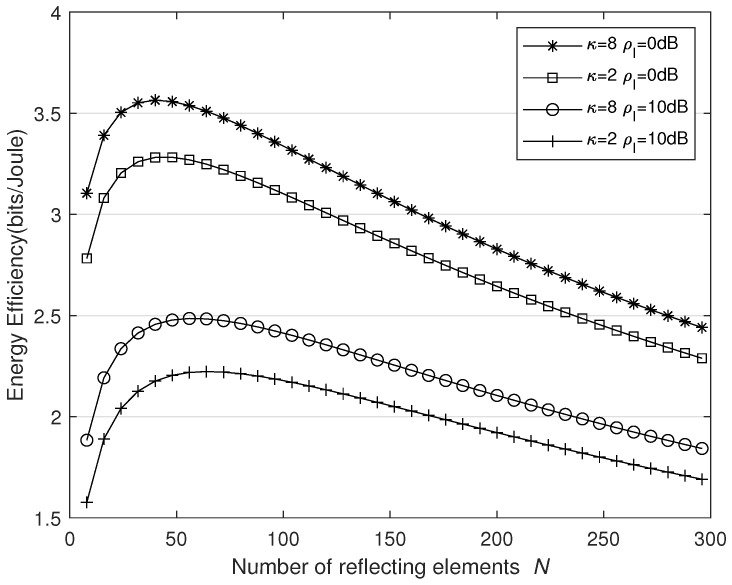
Energy efficiency versus *N* for different κ and $\rho_{I}$ values.

**Figure 8 sensors-25-03756-f008:**
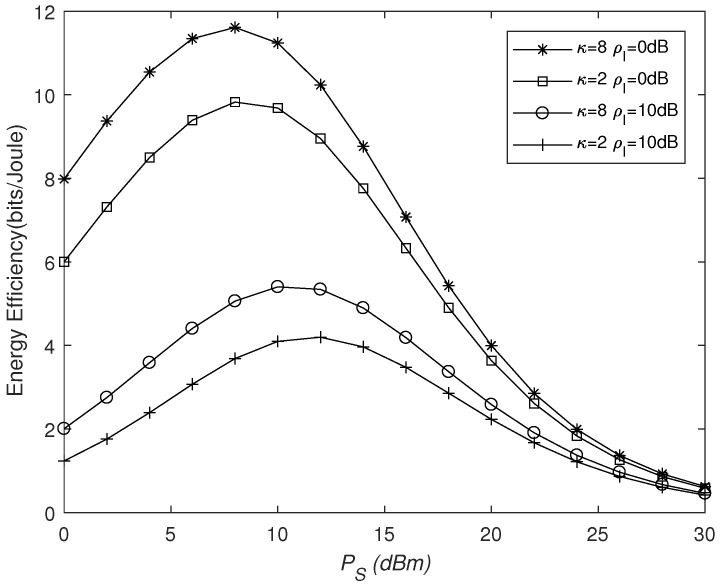
Energy efficiency versus $P_{S}$ for different κ and $\rho_{I}$ values.

**Figure 9 sensors-25-03756-f009:**
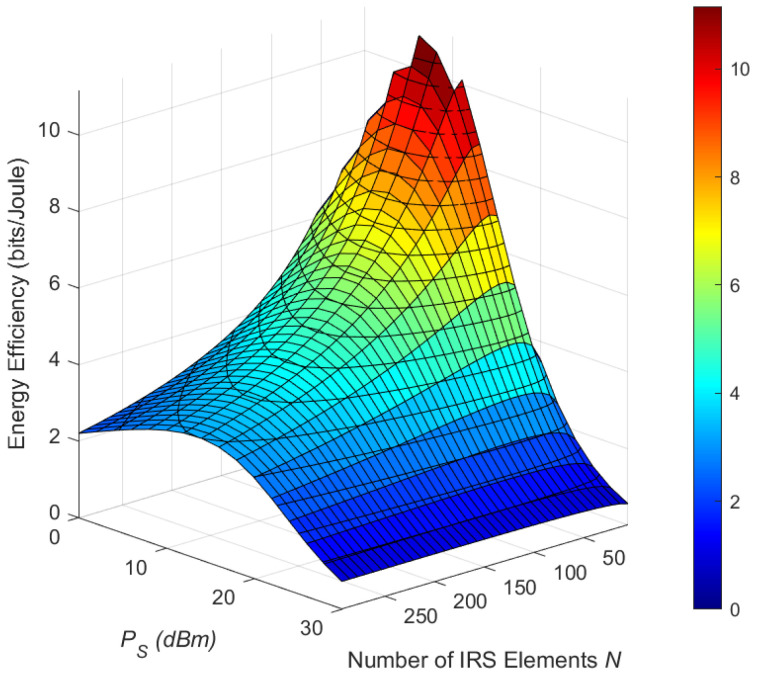
Energy efficiency versus $P_{S}$ and *N* with κ=8 and $\rho_{I}=0$ dB.

**Table 1 sensors-25-03756-t001:** Table of commonly used symbols.

Parameters	Description
*a*, *b*	Non-linear EH model parameters
$\alpha_{SR_{n}}$	Amplitude of S-$R_{n}$ link modeled as CN(0,1)
$\alpha_{R_{n}D}$	Amplitude of $R_{n}$-D link modeled as CN(0,1)
$\alpha_{I_{l}R_{n}}$	Amplitude of $I_{l}$-$R_{n}$ link modeled as CN(0,1)
$\alpha_{\mathrm{SD}}$	Amplitude of S-D link modeled as CN(0,1)
$\theta_{n}$	Phase shift at the *n*th RE of the IRS
$\theta_{{\mathrm{SR}}_{n}}$	Channel phase of $h_{{\mathrm{SR}}_{n}}$
$\theta_{R_{n}D}$	Channel phase of $h_{R_{n}D}$
$\theta_{I_{l}R_{n}}$	Channel phase of $h_{I_{l}R_{n}}$
$d_{\mathrm{SR}}$	Distance between S and the IRS
$d_{\mathrm{RD}}$	Distance between the IRS and D
$d_{\mathrm{SD}}$	Distance between S and D
$d_{I_{l}R}$	Distance between the *l*th interference and the IRS
$h_{\mathrm{SR}}$	Channel vector between S and the IRS (N×1)
$h_{\mathrm{RD}}$	Channel vector between the IRS and D (1×N)
$h_{I_{l}R}$	Channel vector between the *l*th interference and the IRS (N×1)
$h_{\mathrm{SD}}$	Channel of the S to D link
$h_{{\mathrm{SR}}_{n}}$	Channel of the S to $R_{n}$ link
$h_{I_{l}R_{N}}$	Channel of the *l*th interference to $R_{n}$ link
$h_{R_{N}D}$	Channel of the $R_{n}$ to D link
κ	Concentration parameter of the von Mises distribution
*L*	Number of interference sources
*N*	Number of IRS reflecting elements
$n_{d}$	Additive white Gaussian noise (AWGN) (1×1)
η	Energy conversion efficiency
$P_{S}$	Transmit power of source terminal
$P_{I_{l}}$	Transmit power of the *l*th interference source
$P_{\mathrm{Tx}}$	Circuit dissipated power at source terminal
$P_{\mathrm{Rx}}$	Circuit dissipated power at destination terminal
$P_{n}$	Dissipated power at the *n*-th RIS element
$P_{r}$	Average harvested energy
$P_{\mathrm{out}}$	Outage probability
$\rho_{I_{l}}$	Average channel gain of the *l*th interfence-D link
$R_{n}$	IRS reflecting elements
γ	Received SIR of the destination terminal
$\gamma_{\mathrm{th}}$	Threshold value of SINR
S	Source terminal
D	Destination terminal
$I_{l}$	Interfering signals
*v*	Path loss exponent
$w_{\max}$	Circuit dissipated power coefficients at source terminal
φ	Maximum output DC power

## Data Availability

Data will be available upon request to the corresponding author.

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
