# Peer review of "Performance Analysis of an IRS-Assisted SWIPT System with Phase Error and Interference"

_sensors, 2025, doi:10.3390/s25123756_

Round 1

Reviewer 1 Report

Comments and Suggestions for Authors

This paper focuses on the SWIPT system assisted by IRS. Considering the common phase errors and CCI in practice, a system model is established and the system performance is analyzed from the dual perspectives of information transmission and energy acquisition. The accuracy of the theoretical results is verified by simulation, which has certain novelty and practical value. However, the paper still has the following issues:

  1. The literature review section in the first section lists too many literatures, which is somewhat redundant. It is recommended to streamline it appropriately, highlight the research context that is most relevant to this article, and form a clear problem evolution chain.
  2. The paper has numerous issues with formatting and presentation, and a comprehensive revision is necessary. For example, the "most" in line 50 does not have its first letter capitalized; The subject is missing in line 196; Line 202 should be "outage probability", not "outage"; Line 212 is missing "where". Suggest polishing the language throughout the article to enhance the accuracy and professionalism of expression.
  3. The paper assumes that all channels are Rayleigh fading and does not consider the Rician scenario dominated by LOS, which may result in deviations from the actual deployment of IRS. Suggest the author to specify the scope of application of this assumption in the "System Model".
  4. The specific dimensions of multiple channel coefficients (such as hSRn, hRn, hSD) in the paper are not clearly indicated. which is particularly important in systems considering multiple reflection units and multiple interference sources. In addition, the paper does not distinguish between scalars, vectors, and matrices in mathematical symbols, and does not use common bold lowercase letters to represent vectors or bold uppercase letters to represent matrices. It is recommended that the author supplement the dimensional definitions of key variables and standardize the symbol system of the entire text to improve the rigor and readability of the formulas.
  5. The manuscript exhibits inconsistent punctuation usage after equations, including missing commas or periods at the end of multiple formulas and irregular punctuation across different equations. According to academic writing standards, when equations are part of a sentence, appropriate punctuation (commas or periods) must follow the equations to preserve grammatical structure and ensure textual coherence. It is recommended that the authors thoroughly review punctuation placement after all equations in the text and standardize their usage to align with academic rigor and professional presentation.
  6. The nonlinear EH model parameters in Table 1 (such as a=150, b=0.024) need to be explained for their sources.
  7. Check the accuracy of table and figure legends. For example, the symbol "N" in the legend is inconsistent with the main text (not italicized); the parameter labeled "No-line EH" in Table 1 should be corrected to "Non-linear EH"; and Figure 2 displays irregular vertical axis tick marks.
  8. Does the study assume user nodes are stationary? If users are mobile, how should the model be adjusted?
  9. Authors should include the latest references in the field to demonstrate the relevance and timeliness of this research.

Reviewer 2 Report

Comments and Suggestions for Authors

The presented paper is devoted to the already well-known problem related to the use of RIS in future 6G wireless communication. The authors propose to pay special attention to the participation of interfering signals in wireless communication in the presence of propagating phase errors come from RIS structures. The proposed solution is presented in combination with the use of the SWIPT system, for additional recovery of energy from the transmitted signal. The theoretical considerations are supported by formulae that are the basis for numerical simulations. The simulation results are presented in a simple way in graphs. The reviewer believes that the paper can be successfully published after introducing a few minor corrections and extensions:

  1. Discussion should be held on the influence of the Doppler effect on the accuracy of the proposed RIS-SWIPT link analysis method.
  2. In the case of the most effective ranges as a function of S/N and the number of N, it is worth presenting the results in 3D graphs so that one can observe the simultaneous influence of these parameters on the improvement of individual efficiency and effectiveness.
  3. What is the influence of the radio signal carrier frequency and the type of band modulation on the accuracy of the method?
  4. Discussion is necessary on the accuracy of the method in more realistic propagation environments from the NLoS family used in mobile communications.
  5. Other minor comments:
    1. Line 1: it should be „paper”
    2. Line 42: comma instead of dot
    3. Line 50: sentence with a capital letter
    4. Line 82 and next: the mental shortcut should be expanded, i.e. "interference signal"
    5. Line 104: after the comma with a lower case letter
    6. Line 155: IRS cannot generate an optimal phase shift because it is a passive system in the plane of the reflected signal.
    7. Line 234: wasn't this supposed to be chapter 4?
    8. Line 239: the table number should be Arabic
    9. Line 257: missing comma
    10. Figures 5 and 7: missing unit at SNR

Reviewer 3 Report

Comments and Suggestions for Authors

The paper presents the impact analysis of phase errors and interference on IRS systems for SWIPT problems. The work is technically sound and up to date with the state of the art. The authors cover a gap in the literature. However, the paper needs a major improvement in terms of presentation. The first basic issue is the number of variables used. The authors need to present a table with the list of variables so that a reader can quickly reference the table while reading a long paper. The second aspect is that authors try to put more material in Appendixes to make the flow of the paper more dynamic. They already have some aspects in Appendix, but in my opinion other aspects can also be placed in Annex. For example, EE analysis seems a bit long in the main body of the paper. The next point is the use of approximations that have been used in the literature, particularly for the PDF analysis. While i tis valid to use such approximations, the authors need to provide more context. We don’t know in which situations the approximations are valid, or in which range the original reference used the approximation, and if such tightness of the approximation is valid for the problem at hand. For example, the section of system model assumes Rayleigh fading, and we don’t know if this assumption leads to the analysis of the PDF of the error in phases used in the subsequent sections. Therefore, we need more information, even if i tis just a comment on these aspects. Note that i tis also valid to mention that complete accuracy analysis or tightness is still an open problem in the literature, but this statement should be followed by a comment on how it affects to the analysis being presented in the current paper. Finally, and the most important issue identified by this reviewer is the complete detachment of the section of results from the original problem presented in the section of System model. The table of parameters used in the section of results is not linked at all to the system model. We understand that these parameters are relevant for the imprecise phase problem, but we don’t know anything else about distance between IRS and terminals, or operational frequency, noise levels, Tx power transmission, angles (if any) between the IRS and terminals, etc. So, we need the authors to provide a softer transition, and if any of the parameters listed in the table of the section of results has not been described in the previous sections then there is a need of a more explicit explanation in previous sections.

Round 2

Reviewer 3 Report

Comments and Suggestions for Authors

The authors have considerably improved the presentation of the paper. There are only a few details missing. The authors have shown that the approximation of the PDF is accurate. However, the comment was not whether the approximation was accurate or not for the problem at hand. Instead, we just need to understand in which circumstances it is more accurate. Any approximation has a region for more accuracy and another region where the accuracy deviates. For example, the central limit theorem states that the PDF of the sum of arbitrary random variables can be approximated by the PDF of a gaussian random variable, and this approximation is valid when the number of random variables is large. This means that the approximation is less accurate when the number of r.v.s is low. Therefore, all we need in the paper is to briefly understand in which situations the approximation is more accurate, high SNR, high power, or low noise, etc. I believe you have that already in the paper, but it must be further clarified explicitly.

The table of symbols has greatly helped to understand the nomenclature used. However, the idea behind this table is that any reader at any point in the paper can quickly refer to this table and find the symbol under discussion. Unfortunately, when I did this in the section of results, I realized not all symbols of the section of results are included in the table. Specifically, the concentration parameter, and $\rho_I$ or the average interference power are not in the table. Please fix your table but keeping in mind that when a reader goes into the section of results, he might not remember the main variable you described in there for your experiments. So, please include the main variables, which are those that are key for the section of results and to remind a redear of their meaning. It does not have to be the entire universe of variables, please include those that are crucial for the flow of the paper. In the figures of the results, you don’t always use the symbol variable defined in the previous sections, for example outage probability in fig 3 is only mentioned in text and not by its variable. There is another issue with the power Ps in fig. 3. It is defined in dBs, but a power level is in watts and not in dBs, perhaps you refer to power to noise ratio?
